# Topology Informed Surrogate Modeling for Parameter Optimization in Multicellular Models

Andrew Kailiang Jin[*1]
*Georgia Institute of Technology*
Atlanta, USA
ajin40@gatech.edu

Kenji Komiya[*1*2]
*NTT, Inc.*
Kanagawa, Japan
kenji.komiya@ntt.com

Ryo Nishikimi
*NTT, Inc.*
Kanagawa, Japan
ryo.nishikimi@ntt.com

Kunio Kashino
*NTT, Inc.*
Kanagawa, Japan
kunio.kashino@ntt.com

*Abstract*—This study proposes a novel framework to estimate parameters for reproducing target multicellular patterns using an agent-based model (ABM). Two major challenges in multicellular ABMs are estimating cell-level parameters (agent-specific variables) and quantitatively evaluating the topological characteristics of multicellular arrangements under stochastic cell proliferation and death. To address these challenges, we integrate two approaches: Betti vectors and inverse surrogate modeling. The Betti vectors obtained through topological data analysis can consistently represent features of a wide range of multicellular spatial configurations. The inverse surrogate modeling enables direct inference of the corresponding ABM parameters from the target patterns. We validated the proposed framework using zebrafish pigment pattern formation, a representative model of pattern formation driven by multicellular interactions. The results demonstrate that the proposed framework successfully infers ABM parameters. Additionally, when we applied the framework to mutant zebrafish pigment patterns, we estimated parameters with limited similarities to target patterns. This discrepancy suggests that the framework may also serve as a detection tool for identifying missing or unknown mechanisms in the underlying ABM or biological system.

*Index Terms*—Multicellular pattern formation, agent-based model, topological data analysis, surrogate modeling.

## I. INTRODUCTION

**Significance of elucidating mechanisms of multicellular behavior** - Multicellular pattern formation is a biological process in which cells self-organize into spatially distinct structures. This process is essential for the development of complex tissues and organs, as well as during tissue repair and regeneration. These complex patterns arise from the coordination of cell-cell interactions and environmental cues. Many diseases, including some cancers and genetic disorders, yield irregular patterns from abnormal cellular behavior that result in the dysfunction of these multicellular interactions. A goal of biomedical researchers is to understand the mechanisms that control multicellular pattern formation, and to leverage these principles to understand disease pathology and treatments, and to create engineered tissues [1]–[3], artificial organs [4]–[6], and other synthetic biological systems [7]–[9].

**Vitalization of computer simulation of cellular behavior** - Various computational modeling approaches have been developed to understand the mechanisms that underlie multicellular

[*1]These authors contributed equally to this work
[*2]Corresponding author

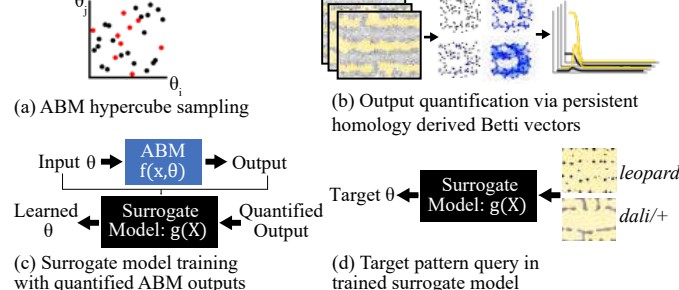

Fig. 1. Proposed agent-based model (ABM) optimization workflow. (a) A training and testing dataset of the desired ABM is generated through hypercube sampling. (b) Cellular ABM outputs can be generically quantified using topological data analysis, resulting in a series of Betti vectors that represent the topological features of the ABM output. (c) Surrogate model training and testing using quantified model outputs (Betti vectors) and known input parameter values. (d) Target patterns can be queried into the trained model to elucidate parameter combinations that may yield similar patterns in simulation.

pattern formation, including reaction-diffusion systems [10]–[13], mechanical models [14]–[16], and agent-based models (ABMs) [17], [18]. Agent-Based Models (ABMs) are multiscale, computational models that represent each cell as an information processing unit or "agent". Each agent is governed by a set of rules that dictate its behavior, which may be dependent on interactions with the simulation environment or other agents. This approach is particularly well-suited for understanding multicellular behavior as a "bottom-up" model because it enables the study of emergent pattern formation through explicit representation of cellular interactions.

**Current challenges for computational modeling of cellular behavior** - Although ABMs are promising for studying multicellular behavior, there are two main challenges for effective and efficient implementation. (See also the excellent survey papers, e.g., [19] and Section 4 in [20].)

- **Difficulty in estimating cellular-level (agent-specific) variables** - One of the typical drawbacks of ABM has long been the difficulty of estimating agent-specific parameters [21]. A common approach to this challenge is to explore a parameter space in a grid search manner [22]. Unfortunately, it is often impractical to explore high-dimensional parameter spaces within a limited time because each ABM simulation of individual cells and their local interactions is computationally intensive.

- **Difficulty in handling stochastic proliferation and death of cells** - In order to handle cell proliferation and death within the ABM framework, it is necessary to have varying model complexity (i.e., number of agents) [20], which significantly affects the difficulty and instability of estimating agent-specific parameters.

Therefore, there has been a continuing demand for methods to resolve these two interdependent issues.

**Our key strategy** - As a method capable of simultaneously addressing these two interdependent issues, we propose a new framework for ABM that combines the principles of *surrogate modeling* and *topological data analysis* (TDA) [23]–[25] (Fig. 1). For the first key feature, we introduce an inverse surrogate module that estimates the parameters of the biological system simulator from the simulator's outcomes, forming biological patterns. For the second key feature, we introduce the TDA-inspired module, which provides a means to assess the global structure of patterns. This allows for meaningful comparison even when the number, position, or scale of features differs between the simulation and the target. We refer to this approach as the Topology-Informed Inverse Parameter Surrogate (TI$^2$PS) framework, emphasizing the integration of both Topology and Inverse parameterization in a surrogate modeling pipeline.

To evaluate the feasibility of the proposed framework, we applied the TI$^2$PS approach to simulations of zebrafish pigment pattern formation in development [26]. This zebrafish pigment system provides an ideal testbed, utilizing two interacting cell types to produce characteristic spatial patterns governed by fundamental cellular behaviors — migration, division, and death [26]. We implemented the inverse surrogate model based on a generalized linear model (GLM) with two different activation functions. Furthermore, we applied the trained inverse surrogate model to estimate parameters from mutant zebrafish pigment patterns.

## II. Related Work

This paper focuses on the intersection of ABMs, surrogate models, and TDA, which are recent popular tools used to shed light on the principles of cellular behavior. In this section, we clarify the novelty of this paper in terms of recent developments regarding these three tools.

**Multiscale extension of ABMs** - Recent reviews have emphasized that ABMs are promising tools for linking macroscopic tissue-level phenomena with microscopic cellular dynamics [20], [27]–[29]. A key challenge in this integration is inferring cell-level behavioral rules from observable tissue-level patterns.

**Affinity of ABMs and surrogate models** - The combination of ABM and surrogate models is an especially active area that has been attracting a lot of attention in recent years [30]–[32]. In general, surrogate modeling aims to use simplified models to approximate ABM's agent-specific variables efficiently. However, a straightforward application to its cellular behavior is not appropriate. Due to the stochastic nature of cell proliferation, death, and other biological processes, the number and arrangement of cells may vary across simulations, making one-to-one correspondence between simulated and reference patterns difficult.

**Affinity of ABMs and TDA** - The combination of topological data analysis with ABM has become popular in recent years [33], [34]. In general, TDA is a set of mathematical techniques that extract structural features (connectivity, loops, and voids) from spatial data [23]–[25]. Instead of relying on point-wise correspondence, TDA provides a means to assess the global structure of patterns, allowing for meaningful comparison even when the number, position, or scale of features differs between the simulation and the target. However, the incorporation of TDA itself does not have the ability to directly alleviate the difficulty of estimating ABM's agent-specific variables.

**Novelty of our approach** - As a nexus of the above three trends in ABM, this paper proposes a new way to make the estimation of cell-level (agent-specific) variables tractable while capturing the global structure of cellular behavior by introducing a surrogate model and a TDA mechanism to ABM in an explicit way.

## III. Method

This section specifies the problem of estimating the parameters of a biological system simulator from observed biological patterns, and describes the proposed inverse surrogate modeling approach to this problem.

### A. Problem Specification

Our goal is to construct the inverse surrogate model that takes as input an observed biological pattern and outputs the parameters of a biological system simulator. The input biological pattern is a set of cell positions $\mathbf{X} = \{\mathbf{X}^c\}_{c \in \mathcal{C}}$ on a two-dimensional plane, where $\mathcal{C}$ is a set of cell types. Each $\mathbf{X}^c = \{\mathbf{x}_i^c\}_{i=1}^{N_c}$ is a set of cell positions of type $c$, where $N_c$ is the number of cells of type $c$ and $\mathbf{x}_i^c \in \mathbb{R}^2$ represents a two-dimensional coordinate of an $i$-th cell of type $c$. The output of our model is simulator's parameters $\boldsymbol{\Theta} = \{\boldsymbol{\theta}^{cc'}\}_{c,c' \in \mathcal{C}}$, where each parameter set $\boldsymbol{\theta}^{cc'} = \{\theta_d^{cc'}\}_{d=1}^{D_\theta} \in \mathbb{R}_{>0}^{D_\theta}$ is indexed by the pair of cell types $c$ and $c'$ to represent the relationships between the same or different cell types.

### B. Betti Vector

The Betti vector can represent the topological features of the cellular point cloud and is derived from the theory of persistent homology, which can capture spatial patterns across multiple scales. For each cell type $c \in \mathcal{C}$, we compute the Betti vector according to the following steps.

**Step 1: Construction of Vietoris-Rips Filtration** - We consider a set of balls with radius $\epsilon/2$, centered at the positions of $M$ cells. Here, $\epsilon \in \mathbb{R}_{\geq 0}$ is called a filtration value, $M$ is the number of cells, and each ball center is indexed sequentially from 1 to $M$. Under this condition, we construct a simplicial complex $X(\epsilon)$, i.e., a set of simplices based on the intersections of these balls. A simplex is defined by a set of center indices, where $k$-simplex consists of $k + 1$ indices

and forms a specific geometric structure. For example, the 0-simplex, 1-simplex, and 2-simplex represent a point, a line segment connecting two points, and a filled triangle connecting three points, respectively. First, we add all indices of ball centers (0-simplices) to $X(\epsilon)$. Next, if two balls intersect, a pair of their indices (1-simplex) is added to $X(\epsilon)$. Finally, if three balls intersect each other, a tuple of their center indices (2-simplex) is added to $X(\epsilon)$.

**Step 2: Computation of Persistent Homology** - We observe the changes that occur in the topological features of $X(\epsilon)$ with the gradual increasing of $\epsilon$. In particular, we focus on the *birth* and *death* of two topological features: a connected component and a loop. The connected component is a 0-degree topological features. At $\epsilon = 0$, each $M$ ball center forms an independent connected component. As the value of $\epsilon$ increases, adjacent components merge to form a larger connected component. The loop is a 1-degree topological feature and is a cycle formed by multiple center points connected with 1-simplices (i.e., line segments). When $\epsilon = 0$, no loops exist. However, as the value of $\epsilon$ increases, new loops emerge, while existing ones may disappear when they are filled by 2-simplices (i.e., triangles). Based on this observation, we can represent the gradual changes in the topological features of each degree $k \in \{0, 1\}$ as

$$\mathcal{P}^k = \left\{ \left( b_j^k, d_j^k \right) \right\}_{j=1}^{N_k}, \quad (1)$$

where $N_k$ is the total number of $k$-degree topological features, $b_j^k$ is the value of $\epsilon$ when $j$-th $k$-degree topological feature appears, and $d_j^k$ is the value of $\epsilon$ when it disappears. $\mathcal{P}^k$ is mathematically derived based on the theory of homology in topology [35].

**Step 3: Computation of Betti curves** - Using the set of birth-death pairs $\mathcal{P}^k$ obtained in Step 2, we define the Betti curves, a function that returns the number of topological features (i.e., connected components and loops) alive at filtration value $\epsilon$ for each dimension $k$, as follows:

$$\beta^k(\epsilon) = \sum_{j=1}^{N_k} \mathbf{1} \left( b_j^k \le \epsilon < d_j^k \right), \quad (2)$$

where $\mathbf{1}(\cdot)$ denotes the indicator function, which equals 1 if the condition given in parenthesis holds and 0 otherwise.

**Step 4: Construction of Betti Vector** - Let $\beta_c^k(\epsilon)$ be the $k$-th Betti curve obtained from the set of cell positions of type $c$, the Betti vector is defined by

$$\mathbf{v}_c^k = \begin{bmatrix} \beta_c^k(\epsilon_1) & \cdots & \beta_c^k(\epsilon_{N_E}) \end{bmatrix}, \quad (3)$$

where $\{\epsilon_i\}_{i=1}^{N_E}$ is a monotonically increasing sequence of filtration values, and $\beta_c^k(\epsilon_i)$ is the Betti number for the filtration value $\epsilon_i$. Finally, by concatenating $\mathbf{v}_c^k$ across all degrees $k$ and all cell types $c$, we obtain the full Betti vector as

$$\mathbf{v} = \begin{bmatrix} \mathbf{v}_{c_1}^0 ; \cdots ; \mathbf{v}_{c_{|\mathcal{C}|}}^0 ; \mathbf{v}_{c_1}^1 ; \cdots ; \mathbf{v}_{c_{|\mathcal{C}|}}^1 \end{bmatrix}^{\mathsf{T}}, \quad (4)$$

where $[\cdot ; \cdot]$ represents the concatenation of row vectors.

This full Betti vector can be used as the input feature for downstream statistical and machine learning analyses. In this study, we use the Betti vector as the input of the proposed inverse surrogate model, where $N_E = 500$, $\epsilon_0 = 0$, and

$\epsilon_i - \epsilon_{i-1} = 0.1\,\mu\text{m}$ for all $i \in \{1, \ldots, N_E\}$. This step size was chosen to balance biological resolution and model input dimensionality. To compute the persistent homology, we used the `ripser` Python library (version 0.6.12) [36].

### C. Inverse Surrogate Model

We use a GLM to estimate the model parameters from the observed biological patterns. The GLM is formulated as follows:

$$\boldsymbol{\theta}^{cc'} = \sigma \left( \mathbf{v}^{\mathsf{T}} \mathbf{W}^{cc'} \right), \quad (5)$$

where $\mathbf{W}^{cc'} \in \mathbb{R}^{D_v \times D_\theta}$ is a matrix of partial regression coefficients corresponding to $\boldsymbol{\theta}^{cc'}$, and $\sigma(\cdot) : \mathbb{R}^{D_\theta} \to \mathbb{R}^{D_\theta}$ is an element-wise nonlinear function. In this study, we use two nonlinear functions: a sigmoid function given by

$$\text{Sigmoid}(x) = \frac{1}{1 + e^{-x}}, \quad (6)$$

and a rectified linear unit (ReLU) function given by

$$\text{ReLU}(x) = \max(0, x) = \begin{cases} x & x > 0, \\ 0 & x \le 0. \end{cases} \quad (7)$$

### D. Optimization

We prepare a dataset containing $N$ sets of cell positions $\mathcal{X} = \{\mathbf{X}_n\}_{n=1}^N$ and optimize the coefficients of the proposed GLM using the Betti vectors $\mathcal{V} = \{\mathbf{v}_n\}_{n=1}^N$, where $\mathbf{v}_n$ is obtained by converting the set of cell positions $\mathbf{X}_n$ with the algorithm described in section III-B. Different optimization methods are used depending on whether the non-linear function of the GLM is a sigmoid function or a ReLU function. For the GLM with a sigmoid function, we optimize the partial regression coefficients based on the Broyden–Fletcher–Goldfarb–Shanno (BFGS) algorithm by minimizing the following sum of squared error:

$$\mathcal{L}_{\text{SSE}} = \frac{1}{N} \sum_{n=1}^N \sum_{c,c' \in \mathcal{C}} \left\| \hat{\boldsymbol{\theta}}_n^{cc'} - \text{Sigmoid}(\mathbf{v}_n^{\mathsf{T}} \mathbf{W}^{cc'}) \right\|_2^2, \quad (8)$$

where $\| \cdot \|_2$ is the $L^2$ norm, and $\hat{\boldsymbol{\theta}}_n^{cc} = \{\hat{\theta}_{nd}^{cc'}\}_{d=1}^{D_\theta} \in \mathbb{R}_{>0}^{D_\theta}$ is a set of ground-truth simulator's parameters corresponding to the $n$-th set of cell positions $\mathbf{X}_n$. For the GLM with a ReLU function, on the other hand, we implement the GLM using a deep neural network tool, and the partial regression coefficients are optimized based on the Adaptive Moment Estimation (Adam) by minimizing the following Huber loss:

$$\mathcal{L}_{\text{Huber}} = \frac{1}{N} \sum_{n=1}^N \sum_{c,c' \in \mathcal{C}} \sum_{d=1}^{D_\theta} \text{Huber}\left( \hat{\theta}_{nd}^{cc'} - \text{ReLU}(\mathbf{v}_n^{\mathsf{T}} \mathbf{W}^{cc'}) \right), \quad (9)$$

where $\text{Huber}(\cdot) : \mathbb{R} \to \mathbb{R}$ is a loss function defined by

$$\text{Huber}(x) = \begin{cases} 0.5x^2 & |x| \le 1, \\ |x| - 0.5 & |x| > 1. \end{cases} \quad (10)$$

## IV. EXPERIMENTS

### A. Data

We create 700 pairs of $\boldsymbol{\Theta}$ and $\mathbf{v}$ for training and 300 pairs for testing our inverse surrogate model using the simulation with an ABM. Since the simulation includes stochastic behavior, we generate five variations of the cell position pattern

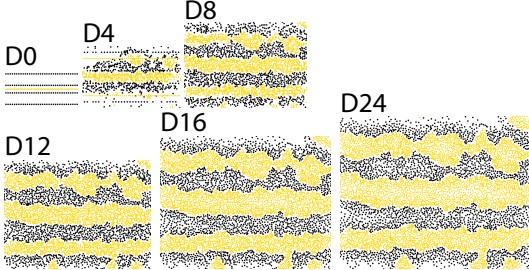

Fig. 2. An example of ABM simulation results. All parameters for the ABM simulation were taken from [26].

$\mathbf{X}$ from single parameter set $\boldsymbol{\Theta}$, and then the average of the Betti vectors calculated from these five patterns is used as the counterpart for that parameter set. In addition, to investigate the performance of the inverse surrogate model, we manually craft examples of cell positions based on known variations in zebrafish pigment patterns, including *dali/+* and *leopard*.

*1) Agent-Based Model for Zebrafish Stripe:* An overview of the simulation process for zebrafish cell positions with ABMs is shown in Algorithm 1 and Fig. 2. The set of cell positions $\mathbf{X} = \{\mathbf{X}^c\}_{c \in \mathcal{C}}$ is derived from an ABM simulation using $\boldsymbol{\Theta}$ generated by Latin Hypercube sampling. The simulation variable $t \in [0, T]$, representing real-world time in days, is introduced for cell position variables: $\mathbf{X}^c(t)$ and $\mathbf{x}_i^c(t)$, where $\mathbf{X}^c(t)$ is a set of cell positions of type $c$ at time $t$, and $\mathbf{x}_i^c(t)$ is an $i$-th cell position of type $c$ at time $t$. We also set $\mathcal{C} = \{\mathrm{B, Y}\}$, where B and Y represent the two types of iridophore relevant for the formation of zebrafish stripes: black melanophores and yellow xanthophores.

We initialize the set of cell positions $\mathbf{X}(0) = \{\mathbf{X}^c(0)\}_{c \in \mathcal{C}}$ by evenly arranging cells in horizontal lines within a simulation domain of width $w_0$ and height $h_0$ as previously described in [26]. Each black cell line contains $N_{\mathrm{B}}$ cells and is positioned $h \in \mathcal{H}^{\mathrm{B}}$ μm from the top edge of the simulation domain, and each yellow cell line contains $N_{\mathrm{Y}}$ cells and is positioned $h \in \mathcal{H}^{\mathrm{Y}}$ μm from the same edge. The lines begin $l_{\mathrm{pad}}$ μm from the left edge and end $l_{\mathrm{pad}}$ μm from the right edge.

Between each time step, i.e., $t \in \mathbb{N} \cup \{0\}$, the simulation domain is stretched horizontally and vertically by $k$ μm to recapitulate the growth of the developing zebrafish. In proportion to the stretch of the simulation domain, the cells are also rearranged as follows:

$$\mathbf{x}_i^c(t) \leftarrow \mathbf{x}_i^c(t) \odot \begin{pmatrix} 1 + \frac{k}{w_t} \\ 1 + \frac{k}{h_t} \end{pmatrix}, \tag{11}$$

where $\odot$ represents the Hadamard product, $w_t = kt + w_0$ and $h_t = kt + h_0$ are functions that return the width and height of the simulation domain at time $t$.

Furthermore, between each timestep, we simulate cell birth and death as a probabilistic event dependent on the neighborhood of each cell location. Cell birth and death increase and decrease the elements of $\mathbf{X}^c(t)$ respectively. To simplify notation, we define the function that counts points within a region on a two-dimensional plane as follows:

$$\#(\mathcal{S}, \mathbf{X}) = \sum_{\mathbf{x} \in \mathbf{X}} \mathbf{1}_{\mathcal{S}}(\mathbf{x}), \tag{12}$$

where $\mathcal{S}$ is a region on a two-dimensional plane, $\mathbf{X}$ is a set of two-dimensional vectors representing the cell positions, and $\mathbf{1}_{\mathcal{S}}(\mathbf{x}) : \mathbb{R}^2 \to \{1, 0\}$ is an indicator function defined by

$$\mathbf{1}_{\mathcal{S}}(\mathbf{x}) = \begin{cases} 1 & \mathbf{x} \in \mathcal{S}, \\ 0 & \mathbf{x} \notin \mathcal{S}. \end{cases} \tag{13}$$

Let $\Omega_{\mathrm{loc}}(\mathbf{x}) = \{\mathbf{x}' \in \mathbb{R}^2 \mid \|\mathbf{x}' - \mathbf{x}\|_2 \le l_{\mathrm{loc}}\}$ be the disk of radius $l_{\mathrm{loc}}$ centered at position $\mathbf{x}$ in a two-dimensional plane and $\Omega_{\mathrm{podia}}(\mathbf{x}) = \{\mathbf{x}' \in \mathbb{R}^2 \mid l_{\mathrm{podia}} \le \|\mathbf{x}' - \mathbf{x}\|_2 \le l_{\mathrm{podia}} + l_{\mathrm{width}}\}$ be the annulus of inner radius $l_{\mathrm{podia}}$ and width $l_{\mathrm{width}}$ centered at position $\mathbf{x}$ in it, the rules for cell death are given as

$$\#\left(\Omega_{\mathrm{loc}}(\mathbf{x}_i^{\mathrm{B}}(t)), \mathbf{X}^{\mathrm{Y}}(t)\right) > \mu \cdot \#\left(\Omega_{\mathrm{loc}}(\mathbf{x}_i^{\mathrm{B}}(t)), \mathbf{X}^{\mathrm{B}}(t)\right)$$
$$\Rightarrow \text{death of } i\text{-th cell of type B}, \tag{14}$$

$$\#\left(\Omega_{\mathrm{loc}}(\mathbf{x}_i^{\mathrm{Y}}(t)), \mathbf{X}^{\mathrm{B}}(t)\right) > \nu \cdot \#\left(\Omega_{\mathrm{loc}}(\mathbf{x}_i^{\mathrm{Y}}(t)), \mathbf{X}^{\mathrm{Y}}(t)\right)$$
$$\Rightarrow \text{death of } i\text{-th cell of type Y}, \tag{15}$$

$$\#\left(\Omega_{\mathrm{podia}}(\mathbf{x}_i^{\mathrm{B}}(t)), \mathbf{X}^{\mathrm{B}}(t)\right) > \xi \cdot \#\left(\Omega_{\mathrm{podia}}(\mathbf{x}_i^{\mathrm{B}}(t)), \mathbf{X}^{\mathrm{Y}}(t)\right)$$
$$\Rightarrow \begin{array}{l} \text{death of } i\text{-th cell of type B} \\ \text{with probability } p_{\mathrm{death}} \text{ per day}, \end{array} \tag{16}$$

where $\mu$, $\nu$, and $\xi$ are hyperparameters obtained from [26].

Let $\Omega_{\mathrm{crowd}}(\mathbf{x}) = \{\mathbf{x}' \in \mathbb{R}^2 \mid \|\mathbf{x}' - \mathbf{x}\|_2 \le l_{\mathrm{crowd}}\}$ be the disk of radius $l_{\mathrm{crowd}}$ centered at $\mathbf{x}$ in the two-dimensional plane, the rules for cell birth are given as

$$\#\left(\Omega_{\mathrm{loc}}(\overline{\mathbf{x}}), \mathbf{X}^{\mathrm{B}}(t)\right) > \alpha \cdot \#\left(\Omega_{\mathrm{loc}}(\overline{\mathbf{x}}), \mathbf{X}^{\mathrm{Y}}(t)\right),$$
$$\#\left(\Omega_{\mathrm{loc}}(\overline{\mathbf{x}}), \mathbf{X}^{\mathrm{Y}}(t)\right) > \beta \cdot \#\left(\Omega_{\mathrm{loc}}(\overline{\mathbf{x}}), \mathbf{X}^{\mathrm{B}}(t)\right), \text{ and}$$
$$\#\left(\Omega_{\mathrm{crowd}}(\overline{\mathbf{x}}), \mathbf{X}^{\mathrm{Y}}(t)\right) + \#\left(\Omega_{\mathrm{crowd}}(\overline{\mathbf{x}}), \mathbf{X}^{\mathrm{B}}(t)\right) < \eta$$
$$\Rightarrow \text{birth of a black cell at } \overline{\mathbf{x}}, \tag{17}$$

$$\#\left(\Omega_{\mathrm{loc}}(\overline{\mathbf{x}}), \mathbf{X}^{\mathrm{Y}}(t)\right) > \phi \cdot \#\left(\Omega_{\mathrm{loc}}(\overline{\mathbf{x}}), \mathbf{X}^{\mathrm{B}}(t)\right),$$
$$\#\left(\Omega_{\mathrm{loc}}(\overline{\mathbf{x}}), \mathbf{X}^{\mathrm{B}}(t)\right) > \psi \cdot \#\left(\Omega_{\mathrm{loc}}(\overline{\mathbf{x}}), \mathbf{X}^{\mathrm{Y}}(t)\right), \text{ and}$$
$$\#\left(\Omega_{\mathrm{crowd}}(\overline{\mathbf{x}}), \mathbf{X}^{\mathrm{Y}}(t)\right) + \#\left(\Omega_{\mathrm{crowd}}(\overline{\mathbf{x}}), \mathbf{X}^{\mathrm{B}}(t)\right) < \kappa$$
$$\Rightarrow \text{birth of a yellow cell at } \overline{\mathbf{x}}, \tag{18}$$

where $\alpha$, $\beta$, $\eta$, $\phi$, $\psi$, and $\kappa$ are hyperparameters whose values are also obtained from [26]. Cell birth can also be a stochastic event, and a melanophore or xanthophore can arise at a candidate location without sufficient neighboring cells with a probability $p^{\mathrm{B}}$ or $p^{\mathrm{Y}}$, respectively.

The cell positions are updated between each timestep by using the Euler method as follows:

$$\mathbf{x}_i^c(t + \delta t) \approx \mathbf{x}_i^c(t) + \frac{d}{dt}\mathbf{x}_i^c(t) \cdot \delta t, \tag{19}$$

where $\delta t$ is a small time step in days, and the ordinary differential equation $d\mathbf{x}_i^c(t)/dt$ is defined by

$$\frac{d}{dt}\mathbf{x}_i^c(t) = -\sum_{\mathbf{x}' \in \mathbf{X}^c(t) \setminus \{\mathbf{x}_i^c\}} \nabla Q^{cc}\left(\mathbf{x}' - \mathbf{x}_i^c(t)\right)$$
$$- \sum_{c' \in \mathcal{C} \setminus \{c\}} \sum_{\mathbf{x}' \in \mathbf{X}^{c'}(t)} \nabla Q^{c'c}\left(\mathbf{x}' - \mathbf{x}_i^c(t)\right). \tag{20}$$

**Algorithm 1:** Simulation of cell positions with ABMs

**Input:** Initial cell positions $\mathbf{X}(0) = \{\mathbf{X}^c(0)\}_{c\in\mathcal{C}}$
**Output:** $\mathbf{X}(T) = \{\mathbf{X}^c(T)\}_{c\in\mathcal{C}}$

1 **while** $t < T$ **do**
2    Stretch the simulation domain and rearrange the cell positions based on (11);
3    Remove the cells based on (14), (15), and (16);
4    Add new cells based on Algorithm 2;
5    $\tau \leftarrow t$;
6    **while** $t < \tau + 1$ **do**
7      Move the cells based on (19) and (22) ;
8      $t \leftarrow t + \delta t$;
9    **if** $t = 4$ **then**
10      Add additional one-layer xanthophore horizontal stripes at 20 and 80% locations;

Here, $Q^{cc'}(\cdot)$ is the Morse potential function given by

$$Q^{cc'}(\mathbf{x}) = R^{cc'} \exp\left\{ -\frac{\|\mathbf{x}\|_2}{r^{cc'}} \right\} - A^{cc'} \exp\left\{ -\frac{\|\mathbf{x}\|_2}{a^{cc'}} \right\}, \quad (21)$$

where $\|\cdot\|_2$ is the $L^2$ norm, and the set of parameters $\boldsymbol{\theta}^{cc'} = \{R^{cc'}, r^{cc'}, A^{cc'}, a^{cc'}\}$, i.e., $D_\theta = 4$, defines the behavior of interactions between the cells of type $c$ and type $c'$. The parameters $R^{cc'}$ and $A^{cc'}$ correspond to the strength scale of repulsion and attraction, and the parameters $r^{cc'}$ and $a^{cc'}$ correspond to the length scale of repulsion and attraction. To prevent the cells from moving outside the simulation domain, we adopt a reflective boundary condition as follows:

$$\mathbf{x}_i^c(t + \delta t) \leftarrow f_{\text{ref}}\left( \mathbf{x}_i^c(t + \delta t), \; \begin{pmatrix} w_{\lfloor t \rfloor} & h_{\lfloor t \rfloor} \end{pmatrix}^\mathsf{T} \right), \quad (22)$$

where $\lfloor \cdot \rfloor : \mathbb{R} \to \mathbb{R}$ is the floor function and $f_{\text{ref}}$ is an element-wise function defined by

$$f_{\text{ref}}(x, L) = L - |x \bmod (2 \times L) - L|, \quad (23)$$

where $\bmod$ is the modulo operation.

As previously described in [26], on the fourth simulated time step, we add a horizontal, single-layer stripe of $N_{\text{add}}$ xanthophores from 20 and 80% from the top of the domain to ensure consistent formation of three stripes when using experimentally determined parameters.

*2) Manually Crafted Zebrafish Stripe:* To validate the effectiveness of our surrogate model, we generated images of simulation targets. Images were created with consideration of known zebrafish pigment patterns, including *dali/+* and *leopard*. Cell positions are manually placed on a domain with dimensions equal to the final domain size of the agent-based model. Cell types are specified during the manual placement of each cell, with a final total cell count of about 6000 cells (*dali/+*: 5451 total cells, 1179 melanophores (m), 4272 xanthophores (x); *leopard*: 6204 total cells, 1029 m, 5175 x).

### B. Parameter Settings

We set the values of the hyperparameters and variables for the ABM-based simulation according to [26]. The width and height of the simulation domain are initialized as $w_0 = 2\,\text{mm}$, and $h_0 = 1\,\text{mm}$, and we extend the domain by $k = 130\,\mu\text{m}$ per day. The parameters for the initial cell positions are set to $\mathcal{H}_\text{B} = \{100, 400, 600, 900\}$ $\mathcal{H}_\text{Y} = \{500\}$, $N_\text{B} = 34$, and

**Algorithm 2:** Simulation of cell birth

**Data:** Maximum number of cells to be born $N_{\text{lim}}$ and that of attempts to simulate cell birth $N_{\text{trial}}$

1 $i \leftarrow 0$, $N_{\text{birth}} \leftarrow 0$;
2 **while** $i < N_{\text{trial}}$ **do**
3    **for** $j \leftarrow 1$ *to* $(N_{\text{lim}} - N_{\text{birth}})$ **do**
4      Sample a candidate position $\overline{\mathbf{x}} \in \mathbb{R}^2$;
5      **if** *there are no cells within $l_{\text{rand}}$ μm of $\overline{\mathbf{x}}$* **then**
6        **if** $\overline{\mathbf{x}}$ *satisfies the condition* (17) **then**
7          Add new black cell at position $\overline{\mathbf{x}}$;
8          $N_{\text{birth}} \leftarrow N_{\text{birth}} + 1$;
9        **else if** $\overline{\mathbf{x}}$ *satisfies the condition* (18) **then**
10          Add new yellow cell at position $\overline{\mathbf{x}}$;
11          $N_{\text{birth}} \leftarrow N_{\text{birth}} + 1$;
12      **else**
13        Sample $q$ uniformly from $[0, 1]$;
14        **if** $q < p^\text{B}$ **then**
15          Add new black cell at position $\overline{\mathbf{x}}$;
16          $N_{\text{birth}} \leftarrow N_{\text{birth}} + 1$;
17        **else if** $q < p^{\text{B+Y}}$ **then**
18          Add new yellow cell at position $\overline{\mathbf{x}}$;
19          $N_{\text{birth}} \leftarrow N_{\text{birth}} + 1$;
20    $i \leftarrow i + 1$;

$N_\text{Y} = 51$. The hyperparameters of the rules for cell death are set to $\mu = 1$, $\nu = 1$, and $\xi = 1.2$, and those for cell birth are set to $\alpha = 1$, $\beta = 3.5$, $\eta = 6$, $\phi = 1.3$, $\psi = 1.2$, and $\kappa = 10$. The radii and width defining the region of the disks or annulus are given as $l_{\text{loc}} = 75\,\mu\text{m}$, $l_{\text{podia}} = 318\,\mu\text{m}$, $l_{\text{width}} = 25\,\mu\text{m}$, $l_{\text{crowd}} = 82\,\mu\text{m}$, and $l_{\text{rand}} = 82\,\mu\text{m}$. The probabilities related to the cell death and birth are set to $p_{\text{death}} = 0.0333$, $p^\text{B} = 0.03$, and $p^\text{Y} = 0.005$. For the simulation of cell birth, we set the maximum number of cells to be born $N_{\text{lim}}$ to 500 and that of attempts to simulate cell birth $N_{\text{trial}}$ to 2. The parameters $\{R^{cc'}, A^{cc'}\}_{c,c'\in\{\text{B,Y}\}}$ are sampled within the range of 0 to 1000, while the parameters $\{r^{cc'}, a^{cc'}\}_{c,c'\in\{\text{B,Y}\}}$ are sampled within the range of 1 to 100. In the ABM simulation of cell development, these sampled values remain unchanged. On the other hand, we use the normalized parameters from 0 to 1 as the ground-truth simulator's parameters. We set the simulation period to $T = 24$ and the small time step to $\delta t = 1$.

The parameters of the Adam optimizer used for training the proposed model are set to $\alpha = 0.001$ (learning rate), $\beta_1 = 0.9$, $\beta_2 = 0.999$, and $\epsilon = 10^{-7}$. The batch size and the number of epochs are 64 and 1000.

### C. Evaluation Metrics

We employed two metrics to evaluate the outcome of the inverse surrogate model. The first is the sum of squared errors (SSE) for the inferred parameters by the surrogate model, which is defined by the equation below.

$$\text{SSE} = \sum_{c,c'\in C} \left( \hat{\boldsymbol{\theta}}^{cc'} - \boldsymbol{\theta}^{cc'} \right)^2, \quad (24)$$

where $\hat{\boldsymbol{\theta}}^{cc'}$ is a true parameter set.

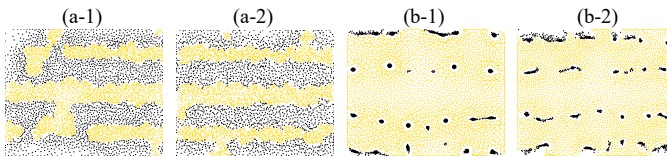

Fig. 3. Validation results with simulated data: (a-1) target stripe pattern and (a-2) simulated pattern with estimated parameters from the target; (b-1) target dot pattern and (b-2) simulated pattern with estimated parameters from the target.

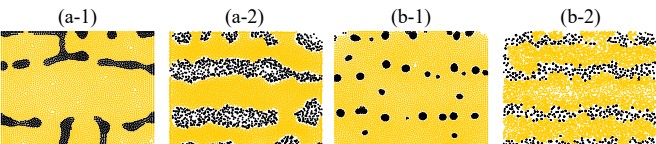

Fig. 4. Validation result with mutant crafted data: (a-1) target distorted stripe pattern of *dali/+* mutant and (a-2) simulated pattern with estimated parameters from the target; (b-1) target polka-dot-like pattern of *leopard* mutant and (b-2) simulated pattern with estimated parameters from the target.

TABLE I
PERFORMANCE COMPARISON WITH WASSERSTEIN DISTANCE

| Model | Sigmoid | ReLU | Random Parameter Selection |
|---|---|---|---|
| SWD | 92.5 | 151.0 | 162.9 |

We also used the Wasserstein distance to evaluate the divergence between the target and simulated Betti vectors. The sum of Wasserstein distance (SWD) across degrees and cell types at time $t$ is defined as:

$$\text{SWD} = \sum_{k \in \{0,1\}} \sum_{c \in \mathcal{C}} \inf_{\gamma \in \Pi(\hat{\mathbf{v}}_c^k(t), \mathbf{v}_c^k(t))} \int_{\mathbb{R} \times \mathbb{R}} (\zeta - \eta)^2 \, d\gamma(\zeta, \eta), \quad (25)$$

where $\hat{\mathbf{v}}_c^k(t)$ and $\mathbf{v}_c^k(t)$ denote the Betti vectors for the target and simulated patterns of degree $k$ and cell type $c$ at time $t$. Here, $\zeta \in \hat{\mathbf{v}}_c^k(t)$ and $\eta \in \mathbf{v}_c^k(t)$ are elements of Betti vectors, and $\Pi(\hat{\mathbf{v}}_c^k(t), \mathbf{v}_c^k(t))$ denotes the set of all valid transport plans $\gamma$ whose marginals match the empirical distributions of Betti vectors from the target and simulated data.

*D. Results*

*1) Validation with Simulated Data:* The surrogate model estimated parameters successfully reproduced spatial patterns equivalent to the target patterns. First, we present examples of the estimation results from our testing set. Fig. 3 (a-1) and (a-2) show the target stripe pattern and the simulation result obtained using the estimated parameters. In both the target and the simulated results, three yellow stripes and two black stripes are clearly visible, demonstrating good agreement between the generated and target patterns. Similarly, Fig. 3 (b-1) and (b-2) display the target dot pattern alongside the simulation result using the corresponding estimated parameters. Both patterns exhibit the same central dot distribution, with laterally elongated melanophore clusters appearing at the top and bottom of the figures. The target dot pattern exhibits a higher cell density compared to the stripe pattern, and the simulation results with inferred parameters reflect this trend. The SSE between the target and simulated patterns was 0.082 for the sigmoid activation function and 0.138 for the ReLU activation function. Table I summarizes the SWD obtained for different approaches to infer parameters. We also evaluated the SWD for multiple simulations using the same parameters. The averaged SWD over those simulations was 13.1, highlighting the robustness of our method of using TDA-based quantification of simulation outcomes to stochastic noise prevalent in ABMs.

*2) Validation with Manually Crafted Data:* We observed some similarities between the target simulation pattern and the surrogate model-guided ABM simulation results. In both ABM simulations, we observe a dominant xanthophore cell population in agreement with the target mutant pigment pattern simulation images (Fig. 4). In the surrogate model guided-ABM *dali/+* pattern, we observe incomplete melanophore stripe formation similar to the target *dali/+* image, but the simulation is unable to recreate the uniform distribution of melanophores within each incomplete stripe. In the surrogate model guided-ABM *leopard* pattern, we observe small polka-dot-like melanophore distributions, but the simulation is unable to produce large polka-dot-like structures that are observed in the target *leopard* pattern. From the estimated parameters for each mutant pattern, a characteristic trend was observed. In both mutant patterns, the model predicts strong long-range repulsion among xanthophores ($R^{xx'}$, $r^{xx'}$), resulting in a predominant xanthophore field in the simulation domain.

## V. DISCUSSION AND CONCLUSION

This study proposed the TI²PS framework based on an inverse surrogate modeling approach for directly estimating parameters in ABMs from spatial cell patterns. For robust quantification of patterns, we implemented the Betti vector, which is a topological descriptor derived from TDA that effectively captures the global structure of cell patterns. While we demonstrated the utility of the framework for pigment pattern formation in zebrafish, the framework is designed to be applicable to diverse biological systems governed by multicellular interactions. This framework provides a foundation for modeling and refining agent-based models of biological systems.

The TI²PS framework successfully estimated simulator's parameters that visually reproduces various pigment patterns of zebrafish. This result highlights several important findings. First, it shows that a learnable mapping exists from ABM simulation results to their corresponding parameters, which is a key advancement in solving inverse problems in ABM-based simulations. Second, it shows that the Betti vector can effectively represent complex cell alignments and function as a practical input feature for the inverse surrogate model. Furthermore, the inverse surrogate model was a simple GLM. This suggests an identifiable relationship between the parameters and the resulting topological features represented by the Betti vector, reinforcing the appropriateness of using Betti vectors to represent ABM output.

Although the ABM simulations using the surrogate model estimated parameters recapitulated some aspects of the target behavior, the model was ultimately unable to fully reproduce

the mutant patterns. There are two possible explanations for this outcome. One possibility is that the parameter combinations required to generate the mutant patterns lie outside the range of the training data. Another possibility is that additional biological mechanisms not captured by the current simulation are involved in generating the mutant patterns. In either case, this framework may serve as a tool to uncover previously unrecognized mechanisms in multicellular interactions.

While the proposed TI$^2$PS framework shows promising performance, some limitations remain. The current study focuses on zebrafish pigment pattern formation, and further validation on other complex biological systems is needed. Improving data generation efficiency, exploring alternatives to Betti vectors, and evaluating surrogate model reliability will help enhance the framework's generality.

## ACKNOWLEDGMENT

We thank Professor Melissa Kemp, Andrew's academic supervisor, for supporting his participation in this research.

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
