# OpenReview forum: "Topology Informed Surrogate Modeling for Parameter Optimization in Multicellular Models"
_IEEE.org/EMBS/BHI/2025/Conference — BHI 2025_

### Official Review · Reviewer_Qmur · 2025-07-15
**An innovative tool for parameter estimation and hypothesis generation**

**Confidence:** 5
**Clarity Of Writing:** excellent
**Clinical Significance:** great
**Methodological Novelty:** excellent
**Overall Rating:** 8

**Experiments And Results:**

great

**Questions For The Authors:**

The validation in this study used simulated data and manually crafted target patterns. What do you foresee as the most important challenge in applying this framework to real-world experimental data, such as 2D or 3D microscopy images, where factors like image noise and cell segmentation errors could impact the stability and reliability of the Betti vectors?

**Strengths:**

The study's main strength lies in its creative combination of agent-based modeling, topological data analysis, and surrogate modeling to tackle a complex biological challenge. This integration allows for an efficient way to infer model parameters from spatial patterns.
It directly addresses the difficult inverse problem in ABMs, recovering the underlying parameters that generate observed outcomes, offering a practical solution that improves the utility of these models in biology.
The framework also enables scientific discovery. When it couldn't fully reproduce mutant zebrafish patterns, this shortcoming pointed to missing mechanisms in the biological model, underlining its potential to reveal gaps in current understanding.

**Summary Of The Paper:**

The authors introduce the Topology-Informed Inverse Parameter Surrogate (TIPS) framework to estimate parameters in agent-based models (ABMs) of multicellular pattern formation. ABMs are valuable for simulating how cells interact to form tissues, but it's challenging to determine which parameters generate a given pattern, especially due to randomness in cell behavior.
TIPS combines topological data analysis with surrogate modeling, and uses persistent homology to translate spatial cell patterns into Betti vectors, which summarize topological features like clusters and loops. A surrogate model then learns to predict the likely ABM parameters that generated those patterns, solving the inverse problem.
The framework was tested on a zebrafish pigmentation model and could accurately recover parameters for simulated stripe and dot patterns. When applied to mutant patterns, it could only partially recreate them, revealing potential limitations in the ABM itself. This suggests TIPS can also help identify missing biological mechanisms, offering a tool for both parameter estimation and model refinement.

**Weaknesses:**

The framework was only tested on zebrafish pigment patterns, so its generalizability to other biological systems remains unproven. Different systems may have dynamics or patterns that are less compatible with the method.
It also relies on a relatively simple surrogate model, a Generalized Linear Model, which may not capture more complex, non-linear relationships found in other biological settings.
The mutant patterns used for validation were manually created rather than based on real images, potentially oversimplifying the problem and limiting the framework’s applicability to real-world data.

---

### Official Review · Reviewer_xzXc · 2025-07-21
**Multicellular model characterization approach using topological data analysis with limited validation and benchmarks**

**Confidence:** 2
**Clarity Of Writing:** great
**Clinical Significance:** fair
**Methodological Novelty:** good
**Overall Rating:** 5
**Final Rating:** 7

**Experiments And Results:**

fair

**Questions For The Authors:**

See above.

**Strengths:**

The paper is well written and easy to follow, even for someone like me who is very unfamiliar with the work in this field.

**Summary Of The Paper:**

The paper describes an approach for characterizing multicellular models through topological data analysis of cell patterns/location and surrogate modeling to derive pattern formation/localization. The approach is validated with a simulated dataset and shows the capability to generate zebrafish patterns with some faithfulness.

**Weaknesses:**

With reservations due to a lack of familiarity in the field, I missed the following when reading through the work.

1. The work misses benchmarks and ablation experiments. The inclusion of TDA and surrogate modeling is presented as novelty. However, there are no benchmarks with existing ABM or alternative reaction-diffusion systems/mechanical models to show the superiority of the proposed method. Similarly, what alternate characterizations beyond Betti vector could be relevant, and how do the choices made justify the approach's efficacy?

2. Is validation with a single cellular pattern type, with two cell types, justified enough to claim the applicability of the proposed method? The paper could have used other references to justify why the chosen pigment pattern for validation suffices to indicate likely utility of the proposed method. Is the two-celltype system the most common/relevant in multicellular simulations?

3. In related work, the authors mention that earlier works have already integrated organ/tissue-level modeling to cellular-level dynamics. It is not clear what the challenges are in integrating the existing 'higher structure' modeling into the proposed cellular dynamics in the paper.

4. The filtration steps are chosen to be 0.1 micrometer. It is not clear, at least to someone not in the field, why and how such choices are justified and generally applicable. Could the paper provide more justification for major parameter/algorithm choices that will likely be relevant for somebody else in the field working on similar problem?


Minor points:

1. Pc is derived from X(ϵ) through mathematically complex operations. ==> This is vague.
2. Sigmoid activations seem to work better, any previous observations like these?

---

### Official Review · Reviewer_YDoC · 2025-07-21
**Topology Informed Surrogate Modeling for Parameter Optimization in Multicellular Models**

**Confidence:** 4
**Clarity Of Writing:** good
**Clinical Significance:** good
**Methodological Novelty:** good
**Overall Rating:** 6

**Experiments And Results:**

good

**Questions For The Authors:**

The paper can be accepeted in IEEE-BHI provided that the following comments are satisfactorily addressed:
- The font size of all texts in Fig. 1 should be the same as the font size of its caption.
- Since a simulation study has been carried out, any type of cells could be simulated including human cells. Please explain why zebra fish cells are chosen. Is it to enable making a comparison with real life experiments?
- Please discuss how your methodology can be used to make more accurate estimation of wall shear stress in developing chicken vasculature simulated in https://doi.org/10.1038/s41598-021-97008-w
- This sentence in the discussion sounds awkward: 'We hope that this framework provides a promising foundation for modeling and tuning multicellular dynamics in biological systems.'

**Strengths:**

- Related work is satisfactorily discussed.
- Methods and the related math are thoroughly discussed.
- The paper is well-written, structurized, and the visualizations are attractive.
- Validation of the proposed ABM model based on pattern formation of zebrafish.

**Summary Of The Paper:**

An agent based model (ABM) is used to estimate parameters for reproduction of target multicellular patterns. Betti vectors and inverse surrogate modeling are merged to estimate cell-level parameters and evaluate topology of mulricellular arrangements. The findings include existence of a learnable mapping from ABN simulation serving as a key advancement in solving inverse problems. Also, effective representation of Betti vector for complex cell alignment has been demostrated.

**Weaknesses:**

- Introduction and related works are structured as bullet points rather than as a storytelling style.
- There is room for more detailed performance evaluation of the proposed model in the form of bar graphs and tables.